# *TaKLU* Plays as a Time Regulator of Leaf Growth via Auxin Signaling

**DOI:** 10.3390/ijms23084219

**Published:** 2022-04-11

**Authors:** Mengdie Zhou, Haixia Peng, Linnan Wu, Mengyao Li, Lijian Guo, Haichao Chen, Baowei Wu, Xiangli Liu, Huixian Zhao, Wenqiang Li, Meng Ma

**Affiliations:** 1College of Life Sciences, Northwest A&F University, Yangling 712100, China; mdzhou0319@163.com (M.Z.); wulinnan@nwafu.edu.cn (L.W.); lmy_925@163.com (M.L.); guolj@nwafu.edu.cn (L.G.); hcchen@nwafu.edu.cn (H.C.); wubaowei250@163.com (B.W.); 1992xlliu@nwafu.edu.cn (X.L.); hxzhao212@nwafu.edu.cn (H.Z.); 2College of Landscape Architecture and Art, Northwest A&F University, Yangling 712100, China; penghaixia@nwafu.edu.cn; 3State Key Laboratory of Crop Stress Biology for Arid Areas, Northwest A&F University, Yangling 712100, China; 4College of Plant Protection, Northwest A&F University, Yangling 712100, China

**Keywords:** wheat, *CYP78A*, biomass, leaf growth, cell division, auxin

## Abstract

The growth of leaves is subject to strict time regulation. Several genes influencing leaf growth have been identified, but little is known about how genes regulate the orderly initiation and growth of leaves. Here, we demonstrate that *TaKLU/TaCYP78A5* contributes to a time regulation mechanism in leaves from initiation to expansion. *TaKLU* encodes the cytochrome P450 CYP78A5, and its homolog *AtKLU* has been described whose deletion is detrimental to organ growth. Our results show that *TaKLU* overexpression increases leaf size and biomass by altering the time of leaf initiation and expansion. *TaKLU*-overexpressing plants have larger leaves with more cells. Further dynamic observations indicate that enlarged wheat leaves have experienced a longer expansion time. Different from *AtKLU* inactivation increases leaf number and initiation rates, *TaKLU* overexpression only smooths the fluctuations of leaf initiation rates by adjusting the initiation time of local leaves, without affecting the overall leaf number and initiation rates. In addition, complementary analyses suggest *TaKLU* is functionally conserved with *AtKLU* in controlling the leaf initiation and size and may involve auxin accumulation. Our results provide a new insight into the time regulation mechanisms of leaf growth in wheat.

## 1. Introduction

Leaves are one of the most important vegetative organs of vascular plants, and provide organic matter for plants through photosynthesis. The time of leaf initial and growth directly determines the characteristics of leaf size, shape, function, etc., which is the key feature of a plant and the basis for the formation of crop biomass and yield [1,2,3]. The growth of leaves from initiation to growth involves a large number of genes and complex physiological processes. Leaves initiate at the flanks of the shoot apical meristem (SAM) in a species-specific chronological order; subsequently, the leaves grow gradually driven by the process of cell division and expansion, which are controlled by the interaction of multiple genes’ expression in time and space [4,5,6,7,8].

The *CYP78A* family (*CYP78As*) is a plant-specific gene family that is highly conserved in land plants [9]. Several members of the *CYP78As* have been implicated in the control of organ growth by promoting cell division and expansion [10,11,12,13,14]. Among them, *AtKLU*/*AtCYP78A5* regulates the growth of leaves, flowers, grains, siliques, and other organs by regulating cell division [15,16,17,18]. For example, *AtKLU* is highly expressed in the inner integument of developing ovules, and determines the growth potential of the seed coat and seed by promoting cell division in *Arabidopsis* [10]. And *AtKLU* inactivity suppresses megasporocyte cell fate and ultimately determines the size and number of seeds in *Arabidopsis* [18]. The *klu* loss-of-function mutants produce smaller leaves and floral organs due to inhibition of cell division [16]. *AtKLU* and *AtCYP78A7* exhibit redundant functions in regulating plastochron by expressing at the edges of meristems in *Arabidopsis*, and the rosette leaves of *cyp78a5 cyp78a7* mutant plants are more and compact than of WT (Wild type) [17,19]. Recently, analyses of multiple mutants revealed that *AtKLU*, *AtCYP78A7* and *AMP1* (*Altered meristem program1*) regulate plastochron length and leaf senescence in the same genetic pathway in *Arabidopsis* [20]. Thus, it can be seen that *KLU* may be involved in the entire life cycle of organ growth from initiation, expansion to senescence, and its growth-promoting effect is precisely regulated in time. However, the molecular mechanism of *CYP78As* controlling organ growth remains elusive.

Recent studies show that *AtKLU* regulates the plastochron in non-cell-autonomous manners, which is just the classic model of hormone action [20]. Metabolome data indicate that *AtKLU* positively regulates leaf senescence by activating cytokinin signaling, and also affects the content of auxin [21]. Moreover, our recent research results reveal that *TaKLU* overexpression promotes auxin accumulation in ovaries [22]. These results confirm the previously hypothesis that the *CYP78As* may generate a hormone-like growth factor to promote organ growth [11,12,23]. However, there is still a huge knowledge gap in our understanding of the relationship between plant hormones and *KLU*. Further, given the lack of positive evidence that *KLU* has a positive effect on leaf and plant growth, it is unclear whether *KLU* could be used to crop improvement.

Wheat is one of the most important food crops, and its yield is of great value to world food security. Despite the orderly initiation and timely growth of leaves will greatly affect leaf characteristics and crop yields, we still relatively know little about how leaf growth is regulated at the time dimension in wheat. Our previous studies show that localized overexpression of *TaKLU* in ovary integument is sufficient to increase grain weight and grain yield per plant in wheat, indicating that *TaKLU* has a great application potential in crop improvement [22]. Here, we demonstrate that *TaKLU* determines the time of leaf initiation and growth, its overexpression is conducive to the increase of leaf size and biomass in wheat. The overexpression of *TaKLU* prolongs the time of leaf expansion, which ultimately leads to an increase in leaf size and biomass. The rescue experiment shows that *TaKLU* and *AtKLU* have conserved functions in regulating leaf initiation and size, and *DR5:GUS* marker detection suggests auxin contributes to this regulating growth mechanism. Thus, we suggest a possible role of *TaKLU* as a time regulator of leaf growth in wheat.

## 2. Results

### 2.1. Overexpression of TaKLU Increased Leaf Size and Biomass by Promoting Cell Division

To demonstrate whether *TaKLU* has a positive effect on promoting leaf growth, two independent transgenic events (*UBI::TaKLU*, including *UBI-1* and *UBI-4* events) with a single copy that have been generated previously [22]. Indeed, the overexpression of *TaKLU* led to a significant increase in the length, width and area of leaves (*p* < 0.05, *n* > 7) (Figure 1A–D), suggesting that the overexpression of *TaKLU* could increase leaf size in wheat. This was complementary to the previous observation that *AtKLU* loss-of-function mutants form smaller leaves [20]. It is well known that the increase of leaf size may affect biomass, so we further investigated the impact of *TaKLU* activity on biomass. *TaKLU*-overexpressing wheat had higher plant height and larger leaves (Figure 1A), which finally led to a significant increase in its biomass (*p* < 0.05, *n* > 16), compared with WT (Figure 1E).

*CYP78As* has been shown previously to regulate flower and grain growth by promoting the proliferation of maternal integument cells [10,11]. However, similar evidence for leaf size is lacking. To determine whether *TaKLU* acts as this maternal factor to promote leaf growth, we investigated the characteristics of leaf epidermal cells of different genotypes of wheat. The results showed that the elongated leaves of *TaKLU*-overexpressing plants had more epidermal cells than those of WT (Figure 1G–J), and there was no significant difference in cell size (*n* > 68) (Figure 1H), indicating that the change in leaf size is due to the increase in cell number. To gain insight into the mechanism of *TaKLU* regulating cell division, we investigated the expression level of several marker genes related to cell wall and cell cycle. Consistent with the overexpression of *TaKLU* leading to a change in the number of leaf epidermal cells in wheat, all of the three cell cycle-related marker genes [24,25] (*CDKA;1* (*CYCLIN-DEPENDENT KINASE 1*), *CYCT1;1* (*CYCLIN T1;1*) and *CDC20* (*CELL DIVISION CYCLE 20*)) in *TaKLU*-overexpressing wheat showed consistent and significant up-regulation (Appendix A), two of the three cell wall metabolism-related marker genes [5,26,27] (*XTH8* (*XYLOGLUCAN ENDOTRANSGLUCOSYLASE/HYDROLASE 8*), *EXPA4* (*EXPANSIN A4*) and *PAE3* (*PECTIN ACETYLESTERASE 3*)) showed differential expression compared with WT in wheat leaves (Appendix A). Thus, the overexpression of *TaKLU* contributed to increasing leaf size by promoting cell division.

### 2.2. TaKLU Overexpression Extended the Time of Leaf Elongation

Our previous study showed that *TaKLU* promotes grain enlargement by prolonging the time of cell division [22]. To explore whether the promoting effect of *TaKLU* on leaf growth is also regulated by time, we conducted a dynamic investigation on the characteristics of the leaf of WT and *TaKLU*-overexpressing plants in wheat. Dynamic morphological observations showed that although the elongation rate of the leaf 5 of *TaKLU*-overexpressing plants was always higher than that of WT, only in the later stages of leaf development (from 8 to 10 days after leaf 5 appearance) was a statistically significant difference (Figure 2A). Correspondingly, the expansion of leaf 5 of WT plants almost stopped at the 8 days after its appearance, while the expansion of the leaf 5 of *TaKLU*-overexpressing plants could be maintained after 10 days after its appearance (Figure 2B,C). Further leaf elongation was prolonged by about two days due to *TaKLU**-* overexpression (Figure 2D). Thus, the increase in leaf size of *TaKLU-*overexpressing plants was attributed to a longer leaf expansion time more than a faster growth rate (Figure 2A–D and Appendix A).

Moreover, in view of the fact that leaf cell division mainly occurs in the division zone at the base of leaf, and *TaKLU* was highly expressed in meristems (Appendix A), we investigated the cell characteristices of division zone to analyze the promoting effect of *TaKLU* on cell division time. Interestingly, cytological observations revealed that the cell division of *TaKLU*-overexpressing leaves may last a longer time than that of WT, even the division zone of the leaf 5 of WT plants basically disappeared on the 16 days after leaf 5 appearance, the division zone of *TaKLU*-overexpressing leaves still existed (Figure 2E). Statistics data showed that the base cells of *TaKLU*-overexpressing leaves were smaller and more round that had a smaller aspect ratio than those of WT at 16 days after its appearance in wheat, and different from the cell morphology in the middle of their leaves (Figure 2E,F). These results implied that the overexpression of *TaKLU* extended the time of cell division in wheat leaves. In summary, these results indicated that *TaKLU* promoted leaf growth by extending the duration of leaf expansion in wheat.

### 2.3. Overexpression of TaKLU Smoothed the Fluctuations of Leaf Initiation Time in Wheat

Given that loss of *AtKLU* function led to an increase in leaf initiation rates due to accelerating cell division around the shoot apical meristems [16,17]. Combining that both *AtKLU* and *TaKLU* were highly expressed in the shoot apical meristem and affected cell division [15] (Appendix A), we speculated that *TaKLU* might also be involved in the regulation of leaf initiation and leaf number in wheat. Thus, to further understand the regulation of the effects of *TaKLU* on leaf development by time, we investigated the initiation times of all leaves at seeding stage. Before this, we first analyzed the grain germination rate of WT and transgenic plants at 24 h after sowing to exclude the effect of germination factors on leaf initiation time. The results showed that the germination time and characteristics of their grains cannot be distinguished throughout the germination stage (Figure 3A,B).

Unlike *klu* mutant plants, which had faster leaf initiation rates and more leaves [17] (Figure 4), both WT and *TaKLU*-overexpressing plants had only 7 leaves and there were no difference in their mean leaf initiation rates of whole growth stages (Figure 3C). However, *TaKLU* overexpressing plants grew slower than WT at the early stages of the seedling (Figure 3D–F), and then the leaf initiation time of the *TaKLU*-overexpressing plants was lower than that of WT (Figure 3G). Interestingly, with the increase in the number of leaves, the leaf initiation rates of all genotype plants was declining, but the decay of the leaf initiation rates of *TaKLU* overexpressing plants was slower than that of WT, even the initial leaf rates of *TaKLU* overexpressing plants was higher than that of WT at the later stage of seedling. (Figure 3G,H). During the whole vegetative stage, *TaKLU*-overexpressing plants appeared to weaken the fluctuation of leaf initiation time, leading to the leaf initiation being gentler (Figure 3C). These results indicated that *TaKLU* overexpression smoothed the fluctuations in leaf initiation by adjusting the initiation time of local leaves, rather than affecting the overall level of the leaf number and initiation rates at the seedling stage.

### 2.4. TaKLU Was Functionally Conserved in Both Monocots and Dicots

To verify whether *TaKLU* and *AtKLU* are functionally conserved in regulating leaf growth, we constructed a vector *pAtKLU::TaKLU* that expressed *TaKLU* driven by the promoter of *AtKLU* (*pAtKLU*) in *Arabidopsis*, and transformed *pAtKLU::TaKLU* vector into *klu* mutants that have been generated previously [28]. Exactly, the phenotype of the increased leaf number, accelerated leaf initiation rates, and small leaves of the *klu* mutants were rescued to normal levels by transferring the *pAtKLU::TaKLU* construct, compared with WT (Figure 4A–D). Moreover, the leaf phenotypes of *pAtKLU::TaKLU;klu* and wild-type plants were almost indistinguishable at the seedling stage (Figure 4A), suggesting that wheat *TaKLU* could rescue the phenotype of *klu* mutants well.

In addition, to further demonstrate whether *TaKLU* has a conserved function in regulating leaf size in both monocots and dicots, we constructed multiple transgenic plants with constitutive promoters (*35S*: *CaMV35S* promoter) driving *TaKLU* expression in *Arabidopsis* (*35S::TaKLU,* including *35S-3*, *35S-5* and *35S-6*) to investigate leaf phenotypes. QRT-PCR detected a significant increase in the expression level of *TaKLU* in all of the transgenic lines (Appendix A). Similar to the phenotype of *TaKLU*-overexpressing wheat, *TaKLU*-overexpressing *Arabidopsis* had the same leaf number and initiation rates, and formed larger leaves, compared with WT (Figure 4E–G). These results suggested that *TaKLU* had a conserved function in regulating leaf growth in both monocots and dicots.

### 2.5. TaKLU Promoted Leaf Growth via Auxin Signaling

Many studies suggest that *KLU* depends on a mobile growth factor downstream to promote organ growth, but there has been a lack of strong molecular evidence linking it to known growth-promoting factors [10,23]. Considering the *klu* mutants showed reduced apical dominance and shortened longevity (Appendix A) [16], and the overexpression of *TaKLU* in wheat led to enhanced apical dominance [22], these phenotypes are classic phenotypes related to auxin [29,30]. We speculated that the regulation effect of *TaKLU* on leaf growth might involve auxin signaling. To test this, we analyzed the accumulation of auxin by using the auxin response reporter *DR5:GUS* in *Arabidopsis*. The strength of the GUS signals suggested that the auxin accumulation in *TaKLU*-overexpressing leaves was increased, while the auxin accumulation of *klu* mutant leaves was decreased than that of WT (Figure 5A–C). Moreover, the constitutive expression of *TaKLU* changed the original accumulation pattern of auxin. Auxin, originally mainly concentrated in the local area as shown in WT, was now distributed accumulated throughout the leaves by constitutively expressing *TaKLU* (Figure 5A–C). To further confirm whether ectopic *TaKLU* expression affects the auxin pathway in wheat and *Arabidopsis*, we tested the expression level of three marker genes related to auxin metabolism and response [31,32]. Compared with WT plants, the expression levels of *YUC 1* (*YUCCA 1*), *ARF 13* (*AUXIN RESPONSE FACTOR 13*), and *PIN1* (*PINFORMED 1*) genes in *TaKLU*-overexpressing plants all changed significantly (Appendix A). These results indicated that *TaKLU* activity affected the accumulation of auxin.

To further understand the relationship between auxin accumulation and the leaf phenotype of different genotypes, we used the auxin analog 1-Naphthaleneacetic acid (NAA) and the auxin synthesis inhibitor 5-methyl-tryptophan (5-MT) to treat different genotypes of *Arabidopsis*. The results showed that the *TaKLU*-overexpressing *Arabidopsis* was more sensitive to NAA treatment, and its leaves were slenderer and even more curled than that of WT, which led to the disappearance of the significant difference between the leaf size of *TaKLU*-overexpressing plants and that of WT (Figure 5D,E). Moreover, although NAA treatment could not completely recover the phenotype of the increased leaf number of the *klu* mutants (Figure 5F), it could improve the phenotype of *klu* mutants, which showed that the petiole length and leaf area of *klu* mutants increased after NAA treatment (Figure 5D,E). In contrast, the leaves of *klu* mutant were smaller and more compact than that of WT after 5-MT treatment, (Figure 5D,E), which was very similar to the leaf phenotype of *cyp78a5 cyp78a7* double-mutants [17], suggesting that the *klu* mutants were more sensitive to 5-MT treatment. These observations indicated that both *TaKLU* and *AtKLU* regulating leaf growth depended on the auxin signaling.

## 3. Discussion

### 3.1. KLU Plays as a Time Regulator of Leaf Growth

Although the leaves of monocot and dicot plants are very different in morphology and structure, the growth of their leaves is both driven by cell division and cell expansion regulating by space and time [34]. *AtKLU* promotes organ growth by stimulating cell division, and the altered *AtKLU* activity can change the time of leaf initiation and senescence [17,21]. Interestingly, our results further revealed that enhanced expression of *TaKLU* prolonged the time of leaf elongation in dynamic observation, which is caused by prolonging the time of cell division (Figure 2). These indicated that *TaKLU* had a time effect on promoting leaf growth.

*CYP78As* have been shown to control organ size by promoting cell division and cell expansion [11,16]. Among them, *OsPLA1*, *AtKLU,* and *AtCYP78A7* are likely homologous genes and reported to act on the initial and subsequent growth of leaves. Their common features include expressing at the periphery of the shoot apical meristem and affecting the development and fate of shoot apical meristem [17,19,35]. For example, *pla1* plants had enlarged shoot apical meristem due to activated cell divisions, which gave rise to a faster leaf initiation rate, and finally formed the doubled number of leaves compared with WT [35]. Based on the expression pattern and its mutant phenotype, *PLA1* was considered to be a timekeeper of plant development [35]. Similarly, inactivation of *AtKLU* accelerated the cell division in the shoot apical meristems in *klu* mutants, which formed a bigger shoot apical meristem to compensate for an increase in leaf initiation rate compared with WT, even *cyp78a5 cyp78a7* double-mutants often die as embryos with supernumerary cotyledon primordial [17]. Combined with the expression of *TaKLU* in young tissues and shoot apical meristem (Appendix A), our work further supports such function of leaf growth time regulation by showing that *TaKLU* overexpression leads to smooth fluctuations in leaf initiation rates and prolonged growth of leaves (Figure 2 and Figure 3).

### 3.2. Relationship between KLU and Plant Hormones

As a monooxygenase, *CYP78As* may play an important role in biosynthesis, especially the synthesis of secondary metabolites such as fatty acids [10,16]. For example, the overexpression of *AtKLU* led to an increase in *Arabidopsis* seed oil content. Transcriptome analysis found that *AtKLU* regulated the expression of several cytochrome P450 genes involved in fatty acid modification [10,16]. Similarly, biochemical analysis showed that CYP78A1 can catalyze the monooxygenase reaction of certain fatty acids [36], and *PLA1* may be involved in the fatty acid synthesis and metabolism pathways required for leaf development [35]. Therefore, it was proposed previously that *CYP78As* were involved in generating a novel signaling factor derived from fatty acids, which can move and promote the growth of cells and organs [10,37]. For *AtKLU*, it has been shown that it promoted growth depending on a mobile growth factor as a traditional hormone in a non-cell autonomous manner [10,16].

Phytohormones are proven to be the determinants of controlling non-cell autonomous growth [38]. However, there has been a controversy about the relationship between *KLU* and hormones. Some researchers believe that *AtKLU* does not directly contribute to biosynthesis or degradation of one of the classical hormones, because of the lack of consistent overlap between the transcriptional responses to *AtKLU* and to phytohormones, and the *klu* mutant’s phenotype cannot be rescued by treatment with exogenous hormones [16]. However, a more recent study showed that *AtKLU* contributed to leaf longevity by activation of cytokinin signaling, because metabolome results showed that the overexpression of *AtKLU* led to increased cytokinin accumulation in leaves [21]. However, changes in auxin accumulation were also detected in their metabolome studies [21]. Here, the expression of the auxin-responsive *DR5:GUS* marker showed *TaKLU*-overexpressing plants and *klu* mutant plants had significant changes in the auxin accumulation in their leaves, and these changes were consistent with *TaKLU/KLU* activity (Figure 5A–C). And our recent research also found that *TaKLU* regulates grain size by affecting the accumulation of auxin [22]. Similarly, in two independent experiments, it was detected that *PLA1* acts on organ development and involved the metabolism of auxin and gibberellin, respectively [39,40]. It is well known that the metabolism and function of different hormones have synergistic and antagonistic effects. Therefore, the changes in the accumulation of multiple hormones have been detected in independent tests related to *AtKLU* and *PLA1*.

Finally, *AtKLU* was known to regulate plant growth by promoting cell division, determining cell fate, and delaying leaf senescence in non-cell autonomous manner [15,16,17,18,21], which were all classic functions of both auxin and cytokinin. In previous experiments, it seemed difficult to determine whether the main factor that *AtKLU* depends on to promote organ development is auxin, cytokinin, or neither. Here, we found that the activity of *KLU* positively regulated the accumulation of auxin in the leaves (Figure 5A–C), and *TaKLU*-overexpressing plants were more sensitive to NAA treatment (Figure 5D), while *klu* mutant plants were more sensitive to the auxin synthesis inhibitor 5-MT treatment (Figure 5D). Thus, we are inclined to support the view that *KLU* regulates leaf development via auxin signaling, but more strong evidence is still needed to prove that *KLU* plays a role in the auxin pathway.

### 3.3. Strategies of TaKLU Applied in Wheat Breeding

Although *TaKLU* appeared to be a potent positive regulator of vegetative organ size (Figure 1), unfortunately, constitutive overexpression did not result in a significant increase in yield due to increased apical dominance of wheat [22]. Combined with the fact that constitutive overexpression of *TaKLU* could change the accumulation and distribution of auxin in leaves (Figure 5A,B), it was reasonable to speculate that constitutive overexpression of *TaKLU* could also alter the accumulation and distribution of auxin in whole plant, thereby changing the plant type of wheat. How to apply *TaKLU* to high-yield crop breeding seemed to be a problem that needed to be constantly explored. Our previous study proposed an effective strategy to locally overexpress *TaKLU* only in the integument of wheat ovary, which not only overcomes the apparent apical dominance of the *TaKLU* overexpressing-plants, but also improved grain weight and grain yield per plant. Similarly, local overexpression of an expansin/TaExpA6 in developing grain resulted in grain enlargement without affecting the number of grains in wheat, which ultimately overcame the opposing relationship between grain size and grain number, resulting in an increase in wheat yield [41]. However, a moderate increase in the size of the vegetative organ was also beneficial to the increase in yield, so we proposed that by overexpressing *TaKLU* in expanding leaves specifically, an increase in leaf size could be achieved without affecting the whole plant type, which may be a possible strategy for *TaKLU* application in wheat improvement.

## 4. Materials and Methods

### 4.1. Plant Materials and Growth Conditions

The *Arabidopsis* materials used in this study were all based on the genetic background of *Arabidopsis thaliana* ecotype Columbia. The *cyp78a5/klu* mutants (Salk_024697C) was obtained from *Arabidopsis* Biological Resources (https://abrc.osu.edu/ accessed on 1 April 2022) and had been identified previously [28]. The transgenic *Arabidopsis* lines and the control materials were grown under the same conditions, which grew under a 16-h/8-h light/dark cycle at 22 °C in the greenhouse, until all of the silique is completely ripe.

The wheat materials used in this study all took JW1 as the genetic background. All of the wheat materials used in this study were grown in the greenhouse, which grew under 24 °C day temperature, 18 °C night temperature and 50% humidity until all of the ears and the leaves were completely yellow.

### 4.2. Vector Construction and Transformation

The *UBI::TaKLU:GFP/GUS* transgenic wheat used in this study was obtained in the early stage of the laboratory [22]. The bar gene was used as a selection marker. The positive transgenic lines were identified based on 0.2% glufosinate (BASTA) through leaf daubing.

To generate overexpression *Arabidopsis* plants, two vectors contained *TaKLU* with different promoters (*CaMV 35S* promoter and the promoter of *AtCYP78A5/AtKLU*) were constructed to transform *Arabidopsis*. The coding region of *TaKLU-A* without stop codon was amplified and cloned into pCAMBIA 3301 vector at *Xba* I/*Bgl* II site under the control of the *CaMV 35S* promoter to generate expression vector p3301-*p35S::TaKLU*. Based on this vector, the promoter of *AtCYP78A5/AtKLU* (AT1G13710) (from –1bp to –2229bp) was amplified from *A. thaliana* genome and replaced the *CaMV 35S* promoter with *Eco*R I/*Xba* I restriction site to construct expression vector p3301-*pKLU::TaKLU*. The primer sequences were listed in Appendix A. These two vectors were introduced into *Arabidopsis* (Columbia-0) by Agrobacterium tumefaciens transformation method as previously described [42]. The transgenic plants were selected by spraying 0.1% glufosinate (BASTA) on seedling leaves. At least three independent transgenic homozygous lines were obtained for each construct.

### 4.3. RNA Isolation and QRT-PCR Analysis

To detect the expression level of *TaKLU*, lateral inflorescences of different genotype *Arabidopsis* plants were collected at 45 days after germination for reverse transcriptase-polymerase chain reaction (RT-PCR) analysis. To detect the expression level of cell cycle (*CDKA;1*, *CYCT;1*, *CDC20*), cell wall (*XTH8*, *EXPA4*, *PAE3*), and auxin-related genes (*YUCCA1*, *PIN1*, *ARF13*), young leaves in the same position from different genotype plants were collected at 10 days after its appearance in wheat and *Arabidopsis*. All samples were frozen in liquid nitrogen and stored at −80°C until RNA extraction.

Total RNA was isolated using the SteadyPure Plant RNA Extraction Kit (Accurate Biotechnology, Changsha, China), and the first-strand cDNA was synthesized via reverse transcription reaction with the *Evo M-MLVRT* Premix Kit (Accurate Biotechnology, Changsha, China). The expression levels of target genes were detected by Quantitative Real-time PCR using the SYBR Green premix *Pro Taq* HS qPCR Kit (Accurate Biotechnology, Changsha, China). Each test has three biologically independent repeats, and the experiment was repeated twice. The results were normalized by the expression levels of internal reference genes *UBC* (for *Arabidopsis*) or *ACTIN* (for wheat). The test results were analyzed with reference to the method of Pfaffl et al. [43]. The primer information used for QRT-PCR detection is shown in Appendix A.

### 4.4. Germination Experiment

We selected transgenic wheat and the control harvested in the same period for germination experiments. The grains of the above genotypes were kept in a moist state and the same environment, and the number of grains germinated which endosperm broken through the seed coat was counted every 12 h. Until all of the grains germinated.

### 4.5. Morphological Analysis

To detect the leaf initiation time of transgenic wheat and the control, we observed the above genotype wheat every day after its germination, and we regard that day when the leaf appeared and the new leaf was visible to the naked eye and appeared about 1–2 cm, as the second day of the appearance of the leaf. The wheat above mentioned was grown in the Hogland nutrient solution according to the instructions.

To dynamically investigate the phenotypic changes of leaves, from the emergence of the fifth wheat leaf (about 50 days after sowing), the length and width of the fifth leaf were measured every other day until the elongation stopped. The length and width of the third and seventh leaves were respectively measured when they were no longer elongated. The leaf area of wheat was calculated by leaf length and width as previously described [44].

In order to explore the conservative function of *TaKLU* and *AtKLU*, we constructed a vector *pAtKLU::TaKLU* and transformed it into homozygous *klu* mutants [28]. The number of leaves of the above genotypes *Arabidopsis* and the control plants was counted at 30 days after its germination and at least 10 plants were measured per line. For accurate measurement, the eighth rosette leaves of different genotypes were photographed with the digital camera, and then their areas were measured with Image J software (https://imagej.en.softonic.com/, accessed on 10 March 2022), at least 5 plants were measured per line. And the images were collected 30 days after germination.

To investigate the phenotypic of leaves of *TaKLU*-overexpressing *Arabidopsis*, the length, width, and area of the eighth rosettes leaves of *Arabidopsis* were measured at 30 days after germination, at least 21 plants were measured per line. The leaf initiation rates in this study was calculated from the ratio of the number of leaves to the growth time (days).

### 4.6. Chemical Treatments

To investigate the relationship between *KLU* and auxin, the 4-week-old *35S::TaKLU Arabidopsis*, *klu* mutants and wild type plants were sprayed with the auxin analogue 1-Naphthaleneacetic acid (NAA) and the auxin synthesis inhibitor 5-methyl-tryptophan (5-MT) (in water), respectively. The spraying concentration of NAA and 5-MT was selected as 0.1 and 500 μM, respectively, according to relevant literature [30,33]. Individual plants at approximately the same developmental stage were selected for treatment. Evenly sprayed IAA, 5-MT, or control solution every 4 days (28, 32, and 36 days after germination) to treat the plants, with at least twenty-four individual plants per line and three lines totally. The observation was made every other day from the first exogenous spray, and final phenotypes were imaged 4 days after the last treatment (40 days after germination). And the images were used to measure the number and area of leaves with Image J software.

### 4.7. Cytological Analysis

In order to determine the sizes of epidermal cells of the leaves, we collected leaf 5 of wheat which appears on 16 days. All of the samples were immersed in carnoy-fixation (Methanol:Glacial acetic acid = 3:1) at 4 °C for 20 min, and then immersed in alcohol at 30%, 50%, 75%, 95%, and 100% in concentration gradient successively until the leaves were decolorized, all of the samples were immersed in each gradient for 30 min. Observed and collected the image of the middle and the bottom part of the leaves through an optical microscope (Olympus, System Microscope BX51, Tokyo, Japan), and finally used the Image J software to measure the cell size. At least 300 cells were measured per line. The total number of epidermal cells per leaf was calculated by dividing the total leaf area by the average cell area.

### 4.8. Histochemical Staining

The *DR5:GUS* line was used as the carrier for in situ analysis of auxins. *DR5:GUS* homozygous was used as the female parent, and the *TaKLU*-overexpressing *Arabidopsis* and *klu* mutant plants were used as the male parent, respectively. Cross and propagate the above parental plants to obtain F_1_ hybrids. The GUS staining assay was performed on the leaves of the F_1_ hybrids at two weeks post-germination, referring to the method reported by Disch et al. [45].

### 4.9. Statistical Analysis

All data obtained in this study were processed in Excel 2013, and an unpaired Student’s *t*-test was used for *p*-values. Significant differences were considered if *p*-values < 0.05, very significant differences were considered if *p*-values < 0.01.

## Figures and Tables

**Figure 1 ijms-23-04219-f001:**
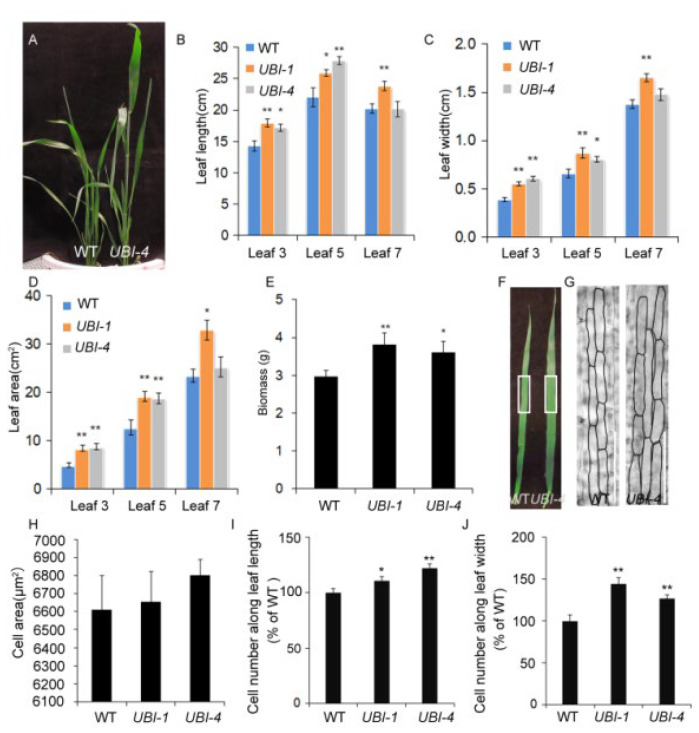
*TaKLU* regulated leaf size and biomass by affecting cell division in wheat. (**A**) WT (Wild type) and *UBI::TaKLU* transgenic wheat lines (*UBI-4*) plants were shown at 80 days after sowing. (**B**–**D**) Statistics on the length (**B**), width (**C**) and area (**D**) of the leaves when they stopped elongating (*n* > 7). (**E**) Statistical analysis of biomass in different genotypes wheat (*n* > 16). (**F**–**J**) Cytological analysis of leaf 5 from WT and *UBI::TaKLU* plants in wheat. (**F**) Leaf 5 of the indicated genotypes at 16 days after its appearance. (**G**) Cell morphology characteristics in the middle of the leaf 5 in panel (**F**) (the position of white rectangle). (**H**) Statistics on the cell area (*n* > 68) of leaf 5 at 16 days after its appearance. (**I**,**J**) Statistics on the cell number along leaf length and leaf width of leaf 5 at 16 days after its appearance (*n* > 9). Bars indicate SE. Size bars represent 200 μm (**G**). Asterisks (*) and (**) indicate significant differences from their WT at *p* < 0.05 and *p* < 0.01 (*t*-test), respectively.

**Figure 2 ijms-23-04219-f002:**
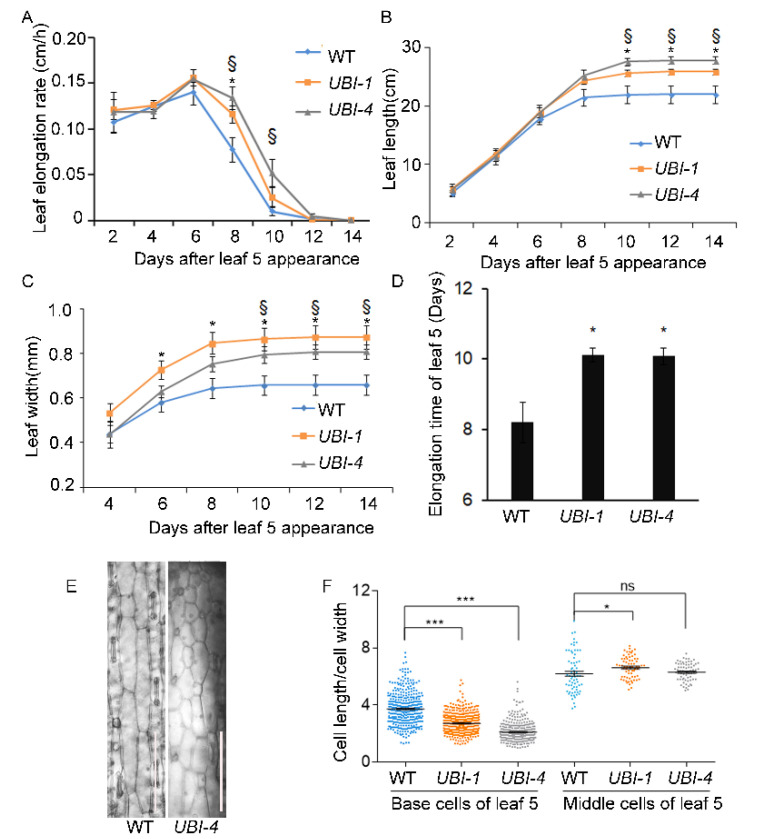
*TaKLU* regulated the time of leaf elongation and cell division. (**A**–**C**) Dynamic analysis of the elongation rate (**A**), length (**B**) and width (**C**) of leaf 5 from 2 to 14 days after its appearance in WT and *UBI::TaKLU* plants (*n* > 7). (**D**) Elongation time of leaf 5 (*n* > 7). Leaf elongation time represents the minimum time required for leaf growth to enter the plateau stage (no significant change in leaf length). (**E**) Characteristics of base cells of leaf 5 were shown at 16 days after its appearance in WT (left) and *UBI::TaKLU* (right, *UBI-4*) plants. (**F**) The aspect ratio of the leaf base cells in panel (**E**) (*n* > 68). Bars indicate SE. Size bars represent 200 μm (**E**). * indicates significant differences between *UBI-1* and WT plants at *p* < 0.05 (*t*-test), § indicates significant differences between *UBI-4* and WT plants at *p* < 0.05 (*t*-test) in panel (**A**–**C**). Asterisks (*) and (***) indicate significant differences from their WT at *p* < 0.05 and *p* < 0.001 (*t*-test) in panel (**D**) and (**F**), respectively. “ns” means no significant difference.

**Figure 3 ijms-23-04219-f003:**
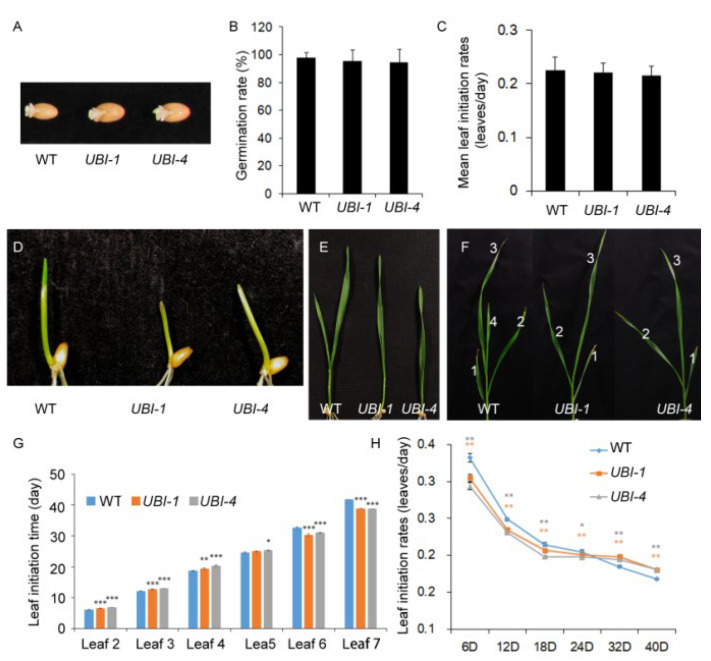
Overexpression of *TaKLU* in wheat influenced leaf initiation time of local leaves. (**A**) Comparison of grain germination between WT and *UBI::TaKLU* plants at 24 h after sowing. (**B**) Statistics of germination rates of WT and *UBI::TaKLU* plants (*n* > 20) at 24 h after sowing. (**C**) Statistics of mean leaf initiation rates of WT and *UBI::TaKLU* plants (*n* > 10). (**D**–**F**) Comparison of growth status between WT and *UBI::TaKLU* plants at the seedling stage (The numbers in subfigure (**F**) indicate the order in which the leaves appeared). (**G**) Dynamic analysis of the leaf initiation time of WT and *UBI::TaKLU* plants (*n* > 10). (**H**) Statistics of leaf initiation rates of WT and *UBI::TaKLU* plants after sowing (*n* > 10). Bars indicate SE. Asterisks (*), (**), and (***) indicate significant differences from their WT at *p* < 0.05, *p* < 0.01, and *p* < 0.001 (*t*-test), respectively.

**Figure 4 ijms-23-04219-f004:**
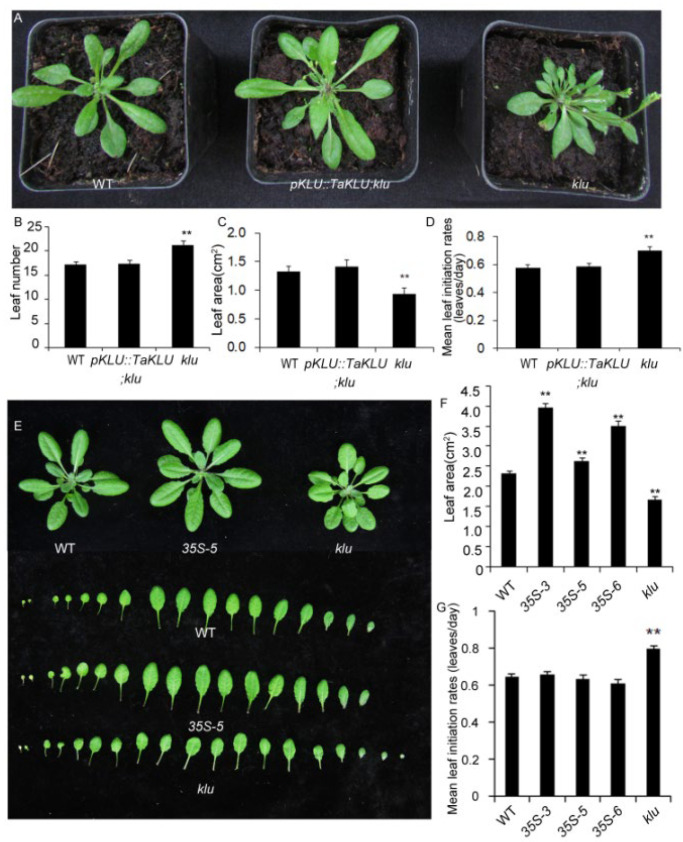
Conserved function of *TaKLU* and *AtKLU* in regulating organ growth. (**A**) *TaKLU* rescued the multi-leaf phenotype of *klu* mutants at 30 days after germination. (**B**–**D**) Quantification of rosette leaf number (**B**), *n* > 10), area (**C**), *n* > 5) and leaf initiation rates (**D**), *n* > 22) in the indicated genotypes at 30 days after germination. (**E**–**G**) Comparison of the leaf morphological characteristics between WT, *35S::TaKLU* (*35S-5*) and *klu* mutant plants at 30 days after germination in *Arabidopsis*. (**E**) The seedling leaves of the indicated genotypes. (**F**–**G**) Statistics on the leaf area (**F**), (*n* > 21) and leaf initiation rate (**G**), (*n* > 13) of WT, *35S::TaKLU* and *klu* mutant plants. Bars indicate SE. Asterisks (**) indicate significant differences from their WT at *p* < 0.01 (*t*-test).

**Figure 5 ijms-23-04219-f005:**
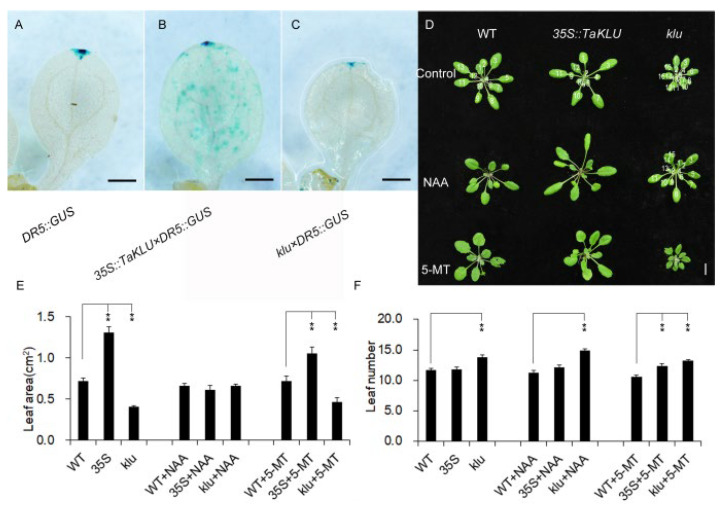
*TaKLU* regulated the accumulation and response of auxin in leaves. (**A**–**C**) *DR5:GUS* marker was used to detect the auxin accumulation in *DR5:GUS* (**A**), *35S::TaKLU* × *DR5:GUS* hybrids (**B**) and *klu* × *DR5:GUS* hybrids (**C**). After resistance screening, the positive hybrids were planted in the same environment with the control groups. Cotyledons of different genotypes *Arabidopsis* were taken for GUS staining at two weeks post-germination. (**D**) Responses of WT, *35S::TaKLU* and *klu* mutant plants to 1-Naphthaleneacetic acid (NAA) and the auxin synthesis inhibitor 5-methyl-tryptophan (5-MT) treatment. Individual plants (*n* > 24) at the same developmental stage were selected for treatment according to previous reports [33]. (**E**,**F**) Statistics on the leaf area (**E**), *n* > 10) and leaf number (**F**), (*n* > 9) of WT, *35S::TaKLU* and *klu* mutant plants. Size bars represent 200 μm (**A**–**C**) and 1 cm (**D**). Bars indicate SE. Asterisks (**) indicate significant differences from their WT at *p* < 0.01 (*t*-test).

## Data Availability

All data can be found in the paper.

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
