# Peer review of "TaKLU Plays as a Time Regulator of Leaf Growth via Auxin Signaling"

_ijms, 2022, doi:10.3390/ijms23084219_

Round 1
Reviewer 1 Report
This manuscript describes the consequences of manipulating the wheat ortholog of the Arabidopsis CYP78A5 (AtKLU), a cytochrome P450 whose deletion was detrimental to organ growth. Here, they overexpressed the TaKLU (wheat ortholog) in wheat under UBQ promoter and Arabidopsis and tested the consequences on leaf initiation, division and growth. OE plants in wheat showed longer leaves similar to the epidermal cells, which led the authors to conclude that more cells were generated. In addition, a comparison of genes involved in cell division showed higher expression, which led them to conclude a positive correlation between TaKLU and increased mitotic activities. Additional comparison made between OE and wt wheat lines showed an extended time of active elongation. Based on the morphology of the leaf's proximal part, the authors related this extension to longer division time.
Similarly, based on a comparison of final leaf length timing, they state the extension of expansion time in OE lines. The leaf initiation rates (#leaves/day) indicated a plateaued change after 18days compared to wild type, but a similar number of final leaves, which is different from similar analysis of the Arabidopsis ortholog. Complementation of the Arabidopsis mutant klu with the wheat gene showed phenotypic rescue and dosage-dependent effect on the leaf sizes. Finally, testing the consequences on the Auxin distribution with DR5:GUS showed accumulation throughout the leaves in constitutively expressing TaKLU plants, and some complementation of the klu mutation with NAA application. These results led authors further to suggest a relationship between KLU activity and auxin signalling.
The manuscript is interesting, and the results are well organized . The rationale for performing the experiments is clear, and the conclusions and discussion are in large in agreement with the results presented.
One caveat is that TaKLU is suggested as a timekeeper both in the title and discussion. However, the results do not fully support the timing characteristic that the authors are trying to emphasize. Timekeeping may be thereby KLU, but additional tests are required to declare that it changes the timing of the cell division and elongation (extending both). Since authors already investigated the expression of cell division genes (Fig. 1S) they could do the same for temporal expression comparison and not satisfy with morphology comparison in the proximal parts (Fig. 2A).
Minor comments:
Row 27—In abstract, the wording of "Unlike" when comparing overexpression of the wheat ortholog and inactivation of the gene in Arabidopsis is inappropriate since these are not same modifications of the gene (overexpression and KO).
Author Response
Response to Reviewer 1 Comments Point 1: This manuscript describes the consequences of manipulating the wheat ortholog of the Arabidopsis CYP78A5 (AtKLU), a cytochrome P450 whose deletion was detrimental to organ growth. Here, they overexpressed the TaKLU (wheat ortholog) in wheat under UBQ promoter and Arabidopsis and tested the consequences on leaf initiation, division and growth. OE plants in wheat showed longer leaves similar to the epidermal cells, which led the authors to conclude that more cells were generated. In addition, a comparison of genes involved in cell division showed higher expression, which led them to conclude a positive correlation between TaKLU and increased mitotic activities. Additional comparison made between OE and wt wheat lines showed an extended time of active elongation. Based on the morphology of the leaf's proximal part, the authors related this extension to longer division time. Similarly, based on a comparison of final leaf length timing, they state the extension of expansion time in OE lines. The leaf initiation rates (#leaves/day) indicated a plateaued change after 18days compared to wild type, but a similar number of final leaves, which is different from similar analysis of the Arabidopsis ortholog. Complementation of the Arabidopsis mutant klu with the wheat gene showed phenotypic rescue and dosage-dependent effect on the leaf sizes. Finally, testing the consequences on the Auxin distribution with DR5:GUS showed accumulation throughout the leaves in constitutively expressing TaKLU plants, and some complementation of the klu mutation with NAA application. These results led authors further to suggest a relationship between KLU activity and auxin signalling. The manuscript is interesting, and the results are well organized . The rationale for performing the experiments is clear, and the conclusions and discussion are in large in agreement with the results presented. Response 1: We appreciated reviewer's positively comment on this study, and we clearly addressed the concerns raised by reviewer in this version of the manuscript. Point 2: One caveat is that TaKLU is suggested as a timekeeper both in the title and discussion. However, the results do not fully support the timing characteristic that the authors are trying to emphasize. Timekeeping may be thereby KLU, but additional tests are required to declare that it changes the timing of the cell division and elongation (extending both). Since authors already investigated the expression of cell division genes (Fig. 1S) they could do the same for temporal expression comparison and not satisfy with morphology comparison in the proximal parts (Fig. 2A). Response 2: Thanks for your rigorous advice and insightful perspective. In fact, in our experiments we collected a large amount of experimental evidence to demonstrate whether TaKLU overexpression extends the time of leaf growth. However, we did not count the specific time of prolonged cell division after proving that the meristem existed for an extended period of time. We will focus on this aspect and do more detailed work in the future research. Following the reviewer's suggestion, we supplemented the data on leaf elongation time to further demonstrate that TaKLU overexpression prolongs leaf growth time (Figure 2D, 2F), and revised the description in the article that TaKLU overexpression extends the time of cell division. Point 3: Minor comments:Row 27—In abstract, the wording of "Unlike" when comparing overexpression of the wheat ortholog and inactivation of the gene in Arabidopsis is inappropriate since these are not same modifications of the gene (overexpression and KO). Response 3: Thanks for your suggestion, we have revised the article according to your suggestion.

Reviewer 2 Report
This work is devoted to the study of the influence of one of the key genes of auxin biosynthesis in plants (CYP78A/KLU) on leaf growth and biomass accumulation in wheat, one of the main food crop. The authors convincingly demonstrated the positive effect of overexpression of the TaKLU gene on the main characteristics of the growth and development of wheat leaves. It has been shown that the main effect of TaKLU overexpression is the accumulation of auxin in plant tissues. The high functional conservatism of the CYP78A/KLU genes in monocots and dicots was confirmed, it was shown that the wheat TaKLU gene in Arabidopsis tissues performs the same functions as its own AtKLU gene and neutralizes the effect of the klu mutation. The authors proposed new approaches for using the TaKLU gene to improve wheat plants and increase its yield. The work is written in an understandable language, well illustrated, will be of a great interest to a wide range of readers, and certainly may be published in IJMS in the current version.
Author Response
Response to Reviewer 2 Comments
Point 1: This work is devoted to the study of the influence of one of the key genes of auxin biosynthesis in plants (CYP78A/KLU) on leaf growth and biomass accumulation in wheat, one of the main food crop. The authors convincingly demonstrated the positive effect of overexpression of the TaKLU gene on the main characteristics of the growth and development of wheat leaves. It has been shown that the main effect of TaKLU overexpression is the accumulation of auxin in plant tissues. The high functional conservatism of the CYP78A/KLU genes in monocots and dicots was confirmed, it was shown that the wheat TaKLU gene in Arabidopsis tissues performs the same functions as its own AtKLU gene and neutralizes the effect of the klu mutation. The authors proposed new approaches for using the TaKLU gene to improve wheat plants and increase its yield. The work is written in an understandable language, well illustrated, will be of a great interest to a wide range of readers, and certainly may be published in IJMS in the current version.
Response 1: We appreciated reviewer's positively comment on this study
Round 2
Reviewer 1 Report
The new revised version of this manuscript took very good care of the small amendments required. Interesting manuscript.